# GraphFramEx: Towards Systematic Evaluation of Explainability Methods for Graph Neural Networks

**Kenza Amara**[1]     **Rex Ying**[2]     **Zitao Zhang**[3]     **Zhichao Han**[3]

**Yinan Shan**[3]     **Ulrik Brandes**[1]     **Sebastian Schemm**[1]     **Ce Zhang**[1]

[1]ETH Zürich, [2]Yale University, [3]eBay China

{kenza.amara@ai, ce.zhang@inf}.ethz.ch, rex.ying@yale.edu

## Abstract

As one of the most popular machine learning models today, graph neural networks (GNNs) have attracted intense interest recently, and so does their explainability. Unfortunately, today's evaluation frameworks for GNN explainability often rely on few inadequate synthetic datasets, leading to conclusions of limited scope due to a lack of complexity in the problem instances. As GNN models are deployed to more mission-critical applications, we are in dire need for a common evaluation protocol of explainability methods of GNNs. In this paper, we propose, to our best knowledge, the first systematic evaluation framework for GNN explainability GRAPHFRAMEX, considering explainability on three different "user needs". We propose a unique metric, the characterization score, which combines the fidelity measures and classifies explanations based on their quality of being sufficient or necessary. We scope ourselves to node classification tasks and compare the most representative techniques in the field of input-level explainability for GNNs. We found that personalized PageRank has the best performance for synthetic benchmarks, but gradient-based methods outperform for tasks with complex graph structure. However, none dominates the others on all evaluation dimensions and there is always a trade-off. We further apply our evaluation protocol in a case study for frauds explanation on eBay transaction graphs to reflect the production environment.

## 1 Introduction

As machine learning models are being deployed to mission critical applications and are having increasingly profound impact on our society, interpreting machine learning models has become crucially important [1, 2]. At the same time, graph neural networks (GNNs) are of growing interest and are ubiquitous in many learning systems across various areas[3–8]). Due to the complex data representation and non-linear transformation, explaining decisions made by GNNs is challenging. The past decade has witnessed the rise of new methods to explain GNN predictions [9–24].

*How do these GNN explanation methods compare with each other? How should we evaluate these GNN explanation methods?* These two questions, unfortunately, are still open today. Today's GNN explainability methods are often evaluated on the inadequate synthetic datasets introduced by [10], later referred as *type 1* (see AppendixA.6 for the types of synthetic data) - where groundtruth is available and often on different grounds — as shown in Table 1. Furthermore, they only consider a small subset of metrics to evaluate their method and this choice is very *different* from method to method. Most papers do not consider the aspect of computing time. They also evaluate their method on an almost accurate GNN model, without considering the influence of GNN accuracy on explainability. As a result, insights obtained in these different papers often *do not reflect their*

K. Amara et al., GraphFramEx: Towards Systematic Evaluation of Explainability Methods for Graph Neural Networks. *Proceedings of the First Learning on Graphs Conference (LoG 2022)*, PMLR 198, Virtual Event, December 9–12, 2022.

**Table 1:** XAI LITERATURE FOR GNN NODE CLASSIFICATION. "Acc" defines the accuracy (F1-score) measured with respect to the groundtruth, "Fid+" and "Fid-" refer to the fidelity metrics as defined in [26] (see Appendix A.4). The column "Time" indicates if the paper has run a comparative analysis of the computation time of the explainability methods. The final column "GNN accuracy" shows if the authors have reported the testing accuracy of their model.

| Paper Type | Year | Explainer | Use type 1 syn data** | Synthetic Acc | Synthetic Fid- | Synthetic Fid+ | Real Acc | Real Fid- | Real Fid+ | Time | GNN Accuracy |
|---|---|---|---|---|---|---|---|---|---|---|---|
| Method [9] | 2019 | LRP | ✓ | ✓ | | | | | | | |
| Method [10] | 2019 | GNNExplainer | ✓ | ✓ | | | | | | | > 0.90 |
| Method [11] | 2020 | PGExplainer | ✓ | ✓ | | | | | | ✓ | 0.92 − 1.00 |
| Method [12] | 2020 | RelEx | ✓ | ✓ | | | | | | | |
| Method [13] | 2020 | PGM-Explainer | ✓ | ✓ | | | ✓ | | | | 0.85 − 1.00 |
| Method [14] | 2021 | RG-Explainer | ✓ | ✓ | | | | | | | |
| Method [15] | 2021 | ZORRO | | | | | | ✓* | | | 0.48 − 0.79 |
| Method [16] | 2021 | SubgraphX | ✓ | | | ✓ | | | | ✓ | 0.86 − 0.99 |
| Method [17] | 2021 | CF-GNNExplainer | ✓ | ✓ | ✓ | | | | | | > 0.87 |
| Method [18] | 2021 | RCExplainer | ✓ | ✓ | | ✓ | | | | ✓ | 0.84 − 0.99 |
| Method [19] | 2021 | Gem | ✓ | | ✓* | | | | | ✓ | |
| Taxonomy [26] (Yuan et al.) | 2020 | GNNExplainer,PGExplainer SubgraphX,DeepLift GNN-LRP,Grad-CAM,XGNN | ✓ | ✓ | ✓ | ✓ | | | | | |
| Taxonomy [25] (Faber et al) | 2021 | Saliency,Occlusion,IntegratedGrad GNNExplainer,PGM-Explainer | | ✓ | | | | | | ✓ | 0.81-1.00 |
| Taxonomy [27] (Li et al) | 2022 | GraphMask GNNExplainer,PGExplainer | | | | | ✓* | | | | |
| Taxonomy [28] (Agarwal et al) | 2022 | VanillaGrad,IntegratedGrad GraphMask,GraphLIME GNNExplainer,PGExplainer PGMExplainer | | | | | ✓* | | | | |

\* Different denomination in the paper, but the same evaluation mechanism as ours.
\*\* Type 1: [10]; Type 2: [25]; Type 3: MUTAG [29], MoleculeNet [30],... See Appendix A.6 for the full synthetic data classification.

*performance on real-world applications!* Most method papers (see upper section of Table 1) have inconsistent rankings when evaluation the methods on type 1 synthetic datasets or on real datasets. Only the taxonomy survey [25] that proposes three novel synthetic benchmarks - *type 2* - has consistent results with real data.

**Evaluation Framework**. In this paper, we aim at overcoming these limitations and propose GRAPH-FRAMEX, the first systematic framework for evaluating explainability methods in the context of node classification. We consider three aspects of *users' needs* in our evaluation protocol. Our framework further distinguishes two types of explanations, according to whether they are *necessary* or *sufficient*. For evaluation, we combine the two fidelity measures, Fid+ and Fid-, that capture the two explanation types, into one single performance metric: the *characterization* score. Our evaluation method does not require groundtruth from synthetic datasets and can be applied to any graph datasets in practice. This paper is the first to study the relation between accuracy and explainability. We evaluate a variety of explainability methods on type 1 synthetic datasets of [10] and ten real datasets. We show the limitations of these specific synthetic datasets. To reflect the production environment, we run a fraud explanation study for eBay transaction graphs. Because runtime is also important, our analysis further compares methods on their average mask computation time. This is also the first paper interested in explaining inaccurate GNN models and the first to investigate the influence of GNN accuracy on the explainer performance.

**Moving Forward**. As an early attempt to systematically investigate evaluation of GNN explainability, this paper also aims to facilitate the assessment of future explainability methods and shed light on how to build more effective explainability methods that would incorporate the advantages of existing methods. We have created an online platform for people to compete and compare their method to a standard leaderboard with our proposed evaluation and a selected set of representative methods. They also have the possibility to integrate their method to the final leaderboard. It also opens new doors to create synthetic datasets that better reflect the complexity of real ones, which we will discuss in

Section 5.2.4. Our code is available at `https://github.com/GraphFramEx/graphframex` and this work is available at `http://www.graphframex.com/`.

## 2 Related work

Confronted to a rapid increase of XAI methods, researchers have tried to identify a list of properties desired of explainable systems and developed concrete tools to help compare and evaluate all of the methods [31, 32]. Following these systematic XAI evaluation reviews, recent studies have proposed to systematically evaluate the performance of explainability methods for GNNs [25–28]. [25] evaluates explainability methods on three new benchmarks for which groundtruth is available to alleviate five pitfalls observed in the widely used type 1 synthetic datasets. But methods are only evaluated with the accuracy metric. Our framework evaluates explainers regardless of the existence of groundtruth. The first attempt to construct an evaluation framework without groundtruth explanations is the paper of Yuan et al. [26]. They evaluate diverse explainability methods on two fidelity scores at different sparsity levels. But simple baselines such as distance and PageRank and gradient-based methods are omitted, while we show their superiority in some settings. [27] adopts the same methodology as [26], but normalizes one of the fidelity scores. Authors of [28] are the first ones to carry out a theoretical study and derive upper bounds on three evaluation metrics: unfaithfulness, instability and fairness mismatch. Like [25], we consider stability and fairness to be optional criteria and not general quality measures. None of the papers studies the relation between accuracy and explainability. Moreover, they do not consider other mask transformation than sparsity.

## 3 Problem setup

Let $G = (\mathcal{V}, \mathcal{E})$ represent the graph with $\mathcal{V} = \{v_1, v_2...v_N\}$ denoting the node set and $\mathcal{E} \subseteq \mathcal{V} \times \mathcal{V}$ as the edge set. Edges may be directed or undirected. The numbers of nodes and edges are denoted by $N$ and $M$, respectively. A graph can be described by an adjacency matrix $\mathbf{A} \in \{0, 1\}^{N \times N}$, with $a_{ij} = 1$ if there is an edge connecting node $i$ and $j$, and $a_{ij} = 0$ otherwise. In addition, nodes in $\mathcal{V}$ are associated with $d$-dimensional features, denoted by $\mathbf{X} \in \mathbb{R}^{N \times d}$.

In the context of node classification, a GNN can be written as a function $f : \mathcal{V} \longrightarrow \mathcal{Y}$, which assigns to nodes in $\mathcal{V}$ labels from a finite set $\mathcal{Y}$. The GNN model is trained with an objective function $\mathcal{L} : \mathcal{Y} \times \mathcal{Y} \to \mathbb{R}$ that computes a cross-entropy loss $s = \mathcal{L}(y, \hat{y})$ by comparing the model's prediction $\hat{y}$ to a ground-truth label $y$. To fuse the information of both node features and graph structure in node representation vectors, GNN models utilize a message passing scheme to aggregate information from neighboring nodes.

Given a pre-trained classifier $f$, our objective is to obtain an explanation model. An "explanation" in the domain of GNNs is a mask or a subgraph of the initial graph, i.e., a set of weighted nodes, edges and possibly node features. The weights on those graph entities relate to their inherent importance for explaining the model outcomes. The explainer model usually performs a feature attribution operation which associates each feature of a computation graph $G_C$ with a weight or relevance score for the classifier's prediction. The computation graph $G_C$ might be the initial graph $G$ or a subgraph around the target node $v_t$ since some methods only look at a k-hop neighbourhood to do predictions. We focus on the contribution of the structural features, namely the edges. To explain each node $v_t$, all the methods compared in this paper generate a mask $\mathbf{M}_E(\mathcal{E}, f, v_t, y_t) \in \mathbb{R}^{|\mathcal{V}| \times |\mathcal{V}|}$, each element of which is the importance score of the edges to the prediction class $y_t$ of the target node $v_t$. The more complex methods also generate a mask $\mathbf{M}_{NF}(\mathcal{V}, f, v_t, c_t)$ on the node features (see Table 5 in Appendix B). At the end, an explanation corresponds to a mask $\mathbf{M}_E$ on the edges and sometimes a mask $\mathbf{M}_{NF}$ on the node features, that operate on the initial graph to form a subgraph $G_S$ with adjacency matrix $\mathbf{A}_S = \mathbf{M}_E \odot \mathbf{A}$ and features $\mathbf{X}_S = \mathbf{M}_{NF} \odot \mathbf{X}$, where $\odot$ denotes elementwise multiplication. We denote by $y_t^{G_S}$ and $y_t^{G_{C \setminus S}}$ the model's predictions for node $v_t$ when taking as input respectively the explanatory or masked graph $G_S$ and its complement or masked-out graph $G_{C \setminus S}$.

**Scope**. Our framework only compares *post-hoc* explainability methods since our focus is on explaining any GNN model. We restricted our study to *input-level* methods because there are currently limited model-level explainability methods [10, 20]. We evaluate both *model-aware* and *model-*

*agnostic* methods in the context of node classification tasks. See Appendix A for the full definitions and taxonomy.

# 4 Method

This section presents the three design choices made by the users and the evaluation metrics used to assess explainers performance.

## 4.1 Multi-objectives for explainability

To build GRAPHFRAMEX, we start from the perspective of the data subject. Users design the framework based on their expectations on the produced explanations. They can make choices on three dimensions: the explanation focus, the mask nature and the mask transformation strategy.

**Aspect 1: the focus of explanation.** Some users want to explain why a certain decision has been returned for a particular input. In this case, the task of explaining has a more applied nature: they are interested in the *phenomenon* itself and try to reveal findings in the data, i.e. explain the true labeling of the nodes. The model's predictions are ignored in the explanation process. Others prefer to explain how the model works. In this case, they are interested in the GNN *model* behavior and try to explain the logic behind the model, i.e. the predicted labels. These equally complementary and important reasons demand different analysis methods. The choice of explanation focus determines the explanation objective and evaluation.

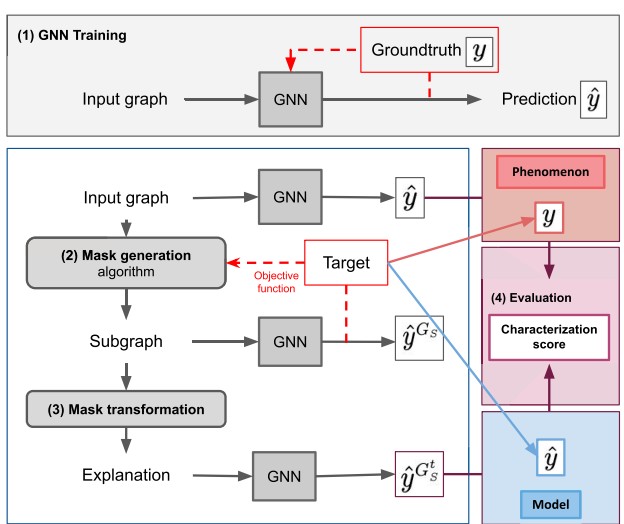

**Figure 1:** General protocol. The explanation focus is the **phenomenon** or the **model**. (1) A GNN model learns to predict the label $\hat{y}$ of each node in the input graph. For the explanation of node labels (true or predicted), we use this pre-trained model. The explainability method generates a soft mask $M_E$, which operates on the input graph to return a subgraph $G_S$. (2) The goal is to reproduce a target label: $y$ or $\hat{y}$. (3) The mask is transformed to output the final explanatory subgraph $G_S^t$. (4) We evaluate $G_S^t$ by comparing its predicted label to our target.

**Aspect 2: mask nature: hard or soft mask.** Edge masks $\mathbf{M}_E$ are normalized so that each weight lies between 0 and 1. To convey human-intelligible explanation, we can directly operate the initial *soft mask*, $\mathbf{M}_E^{soft} \in [0,1]^{M \times M}$ on $G_C$ and return an explanatory subgraph $G_S^{soft}$, where the edge weights reflect the relative importance of edges. But, users might prefer a non-weighted subgraph $G_S^{hard}$ as explanation. In this case, once the mask has been transformed (Aspect 3), we convert the mask into a *hard mask*, $\mathbf{M}_E^{hard} \in \{0,1\}^{M \times M}$ by setting every positive values to 1.

**Aspect 3: the mask transformation.** Because there is no such thing as a "good" size for an explanation, it is even harder to compare explainability methods. Existing explainability methods return different sizes of explanations by default. To make them comparable, most papers propose to fix a sparsity level to apply to all explanations and compare the same-sized explanations [16, 18, 33]. We define three strategies to reduce explanation size: sparsity, threshold and topk (see Appendix B), which transform the edge mask $M_E$ into a sparser version $M_E^t$. We decide to use the topk strategy because it is the only strategy that enforces a maximum number $k$ of edges independently of the size of the graph and the explainer methodology. This independence property is important as human-intelligible explanations cannot exceed a certain number of graph entities. Too small explanations omit important elements and will not be sufficient, while too big explanations contain irrelevant nodes and edges and will not be necessary.

## 4.2 Evaluation

**Phenomenon**

$$fid_+ = \frac{1}{N} \sum_{i=1}^{N} \left| \mathbb{1}(\hat{y}_i = y_i) - \mathbb{1}(\hat{y}_i^{G_{C \setminus S}} = y_i) \right|$$

$$fid_- = \frac{1}{N} \sum_{i=1}^{N} \left| \mathbb{1}(\hat{y}_i = y_i) - \mathbb{1}(\hat{y}_i^{G_S} = y_i) \right|$$

**Model**

$$fid_+ = 1 - \frac{1}{N} \sum_{i=1}^{N} \mathbb{1}(\hat{y}_i^{G_{C \setminus S}} = \hat{y}_i)$$

$$fid_- = 1 - \frac{1}{N} \sum_{i=1}^{N} \mathbb{1}(\hat{y}_i^{G_S} = \hat{y}_i)$$

We define multiple dimensions on which we can evaluate explanations. If we have the ground-truth explanations, we can use the accuracy metric. In most of the cases, ground-truth explanations are unknown and explanatory subgraphs are evaluated on their contribution to the initial prediction.

**Fidelity**. To be independent from any ground-truth explanations, we suggest using the fidelity measures. We extend the definitions in [26] by considering in addition the explanation focus. We make some adjustments: for the phenomenon focus, the fidelity is measured with respect to the ground-truth node label $y$; for the model focus, it is measured with respect to the outcome of the GNN model $\hat{y}$. In the context of node classification, the indicator function certifies whether the predicted class of a subgraph corresponds to the desired class defined as the true label $y$ in the phenomenon focus or the predicted label for the whole graph $\hat{y}$ in the model focus.

**Typology**. Considering the large spectrum of possible explanations, we propose to classify explanations in two categories based on their fidelity scores. Each category defines the role of the explanation in producing the observed outputs: the explanation can be necessary and/or sufficient.

- SUFFICIENT EXPLANATION An explanation is sufficient if it leads by its own to the initial prediction of the model explanation. Since other configurations in the graph may also lead to the same prediction, it is possible to have multiple sufficient explanations for the same prediction. A sufficient explanation has a $fid_-$ score close to 0. We later report $(1 - fid_-)$ in our experiments.
- NECESSARY EXPLANATION An explanation is necessary if the model prediction changes when you remove it from the initial graph. Necessary explanations are similar to counterfactual explanations [34]. A necessary explanation has a $fid_+$ score close to 1.

An explanation is a characterization of the prediction if it is both necessary and sufficient. It can be interpreted as the certificate for a specific class or label. Explainability methods should aim at returning this type of explanations as they are the most informative and complete.

**General performance metrics**. A variety of functions exists to combine Fidelity+ and Fidelity-measures into a single metric on the overall quality of the explanation such as the area under Fid+/(1-Fid-) curve (AUC). For users interested in only one aspect of an explanation, *i.e.* necessary or sufficient, we suggest to use the fidelity scores independently, *i.e.* Fid- or Fid+, and compare the performance of explainability methods with Fid+@K or (1-Fid-)@K metrics.

**Characterization score**. In this paper, we recommend the characterization score as a global evaluation metric, due to its ability to balance the sufficiency and necessity requirements. This approach is analogous to combining precision and recall in the Micro-F1 metric. The *charact* score is the *weighted harmonic mean* of Fid+ and 1-Fid- as defined in Equation 1:

$$charact = \frac{w_+ + w_-}{\frac{w_+}{fid_+} + \frac{w_-}{1 - fid_-}} = \frac{(w_+ + w_-) \times fid_+ \times (1 - fid_-)}{w_+ \cdot (1 - fid_-) + w_- \cdot fid_+} \tag{1}$$

where $w_+, w_- \in [0, 1]$ are respectively weights for $fid+$ and $1 - fid_-$ and satisfy $w_+ + w_- = 1$. In the context of explainability, it is important to know that the explanation is leading to the prediction, i.e. sufficient, but also essential for this output, i.e. necessary. As seen in Equation 1 and Fig. 2, the characterization score with equal weights on Fid+ and (1-Fid-) is low as soon as one of the two terms is low. It reflects the strong simultaneous dependency of the characterization score to both fidelity measures. In addition, it is possible to vary the weights $w_+$ and $w_-$ to

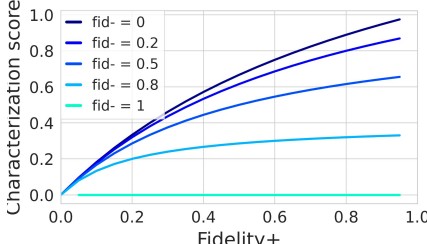

**Figure 2:** Characterization score for $w_+ = w_- = 0.5$

compare explainers more on one aspect rather than the
other.

**Efficiency**. Efficiency relates to the trade-off between performance, assessed by the characterization score, and computation time of an explanation. A method is very efficient if it quickly generates explanations that well characterize the GNN predictions. This is an important criteria as users might want rapid answers to their why-questions.

## 5 Results

We evaluate existing methods on their efficiency, characterization power, and type of explanations. No method is dominating the others in all aspects. We also discuss here the limitations of previous evaluation protocols.

### 5.1 Experimental settings

We describe the setup and implementation details for the explainability procedure. See Appendix B for more details on the datasets statistics, the methods and the experimental protocol.

**Datasets**.

- **Synthetic datasets** We use type 1 synthetic datasets introduced by [10]. We refer the reader to Appendix A.6 to learn more about the 3 classes of existing synthetic datasets in explainability for GNNs. Ground truth explanations are available.
- **Real datasets** We use 10 publicly available datasets to evaluate our framework on real graphs: the citation network datasets [35], the Facebook Page-Page network dataset [36], the actor-only induced subgraph of the film-director-actor-writer network [37], the WebKB datasets [37], and the Wikipedia networks [36]. We use the code accessible in Pytorch geometric.
- **eBay** We test our evaluation framework on a real-world eBay transaction graph dataset. This is a binary node classification task where transaction nodes are labeled as legit or fraudulent. The objective is to explain fraudulent nodes. The eBay graph dataset is a very large sampled real-world dataset with 289k nodes (208k transaction nodes) and 1% of all nodes (1.48% of transaction nodes) are fraudulent. This is a typical example of a rare event detection task.

**GNN models**. By default, we use the graph convolutional networks (GCN) [38]. Besides GCN, we also evaluate explainability methods on graph attention networks (GAT) [39] and graph isomorphism networks (GIN) [40]. Results using GAT and GIN models are presented in Appendix C.

**Explainers**. To explain the decisions made by the GNNs, we adopt different classes of explainers including structure-based methods, gradient/feature-based methods and perturbation-based methods. We refer the reader to Appendix A.3 for the full taxonomy and to Appendix B.2 for more details on the explainability methods. In our experiments, we compare the following methods: **Random** gives every edge and node feature a random value between 0 and 1; **Distance** assigns higher importance to edges that have lower distance to the target node; **PageRank** measures the importance of edges following the personalized PageRank strategy with automatic restart on the target node [41, 42]; **Saliency (SA)** measures node importance as the weight on every node after computing the gradient of the output with respect to node features [9]; **Integrated Gradient (IG)** avoids the saturation problem of the gradient-based method Saliency by accumulating gradients over the path from a baseline input (zero-vector) and the input at hand [43]; **Grad-CAM** is a generalization of class activation maps (CAM) [44]; **Occlusion** attributes the importance of an edge as the difference of the model initial prediction prediction on the graph after removing this edge [25]; **GNNExplainer** computes the importance of graph entities (node/edge/node feature) using the mutual information [10]; We also try **Basic GNNExplainer** that considers only edge importance; **PGExplainer** is very similar to GNNExplainer, but generates explanations only for the graph structure (nodes/edges) using the reparameterization trick to overcome computation intractability [11]; **PGM-Explainer** perturbs the input and uses probabilistic graphical models to find the dependencies between the nodes and the output [13]; and **SubgraphX** explores possible explanatory subgraphs with Monte Carlo Tree Search and assigns them a score using the Shapley value [16].

**Protocol**. In this work, we focus on node classification tasks and compare local, that is input-level, explainability methods. We train one of the three GNN models. Once trained, we use the GNN to do predictions on a testing set. Explanations are then eventually transformed with the topk strategy. We

evaluate the methods with the fidelity measures and the characterization score with equal weights $w_+ = w_- = 0.5$ in four different settings defined as the combinations of the two possible focus, *phenomenon* and *model*, and mask nature, *hard* or *soft* masks.

## 5.2 Main results

### 5.2.1 Explainer efficiency and type of explanation on real datasets

The legend of figure 3 shows the overall ranking of each explainability method. We rank them on their characterization score averaged on all real datasets for explanations of size 10 edges in the four settings (*phenomenon* / *model*, *hard* / *soft* mask). Saliency has the highest overall characterization score. More generally, gradient/feature-based methods are better than perturbation-based methods.

The overall characterization score of the twelve explainers on the real datasets is also evaluated against their average computation time of an explanatory mask. Left plot of Figure 3 shows that, in addition to having the best characterization score, Saliency is also the most efficient. In the setting where we explain the model with a hard mask, we observe that Occlusion has the best overall score but is $10^4$ times slower than Saliency.

We compare the methods on the type of explanation they return. On the right plot of Figure 3, methods scoring high on the x-axis return necessary explanations, while those scoring high on the y-axis return good sufficient explanations. We observe that Saliency is by far the best one to return necessary explanations. But, for sufficient explanations, Occlusion, Grad-CAM and PageRank are better choices. As a general remark, we observe that most of the methods are able to return very good sufficient explanations as their explanations have a fidelity- score close to 0. But very few generate necessary explanations: only Saliency, Distance and Occlusion reach a fidelity+ score greater than 0.6 in at least one of the four settings.

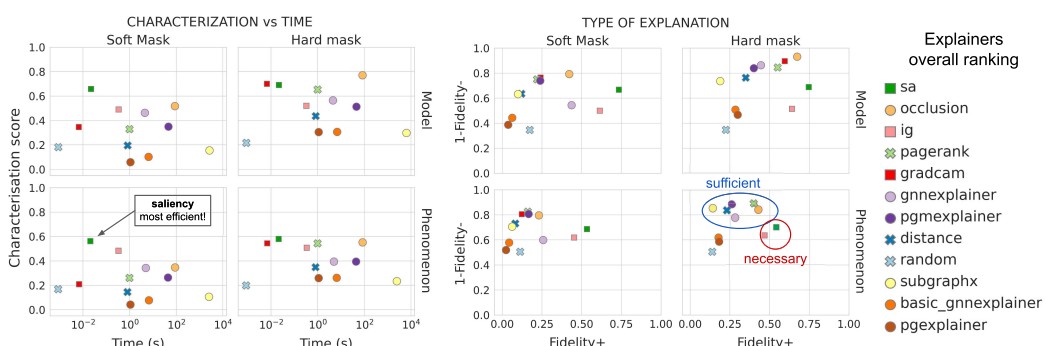

**Figure 3:** Results on real datasets. (left) Performance and computation time. (right) Type of explanation returned by each explainability method. *sa* - Saliency. *ig* - Integrated Gradient.

### 5.2.2 Explaining wrong predictions

Most of the papers report GNN testing accuracy greater than 80% and all of them test their explainers on a mixture of correct and wrong predictions (see Table 1). But when ignoring this distinction, they unknowingly take a different focus. When they explain correct predictions, they target the true label and explain both the phenomenon and the GNN model. When they explain wrong predictions, the predictions by the GNN do not correspond to the true label and, therefore, they can only get an insight of the GNN logic. We decide to study what happens to our explainers ranking if we

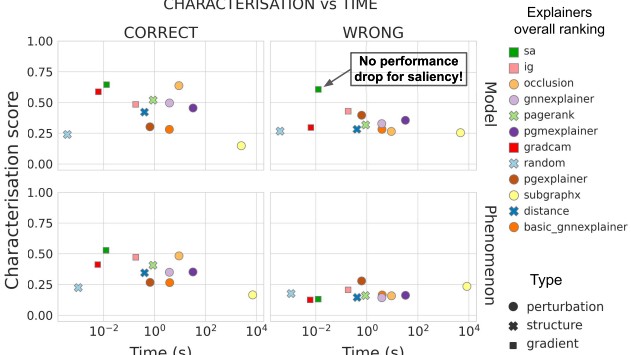

**Figure 4:** Average performance when explaining only correct (left) or only wrong (right) predictions on 5 real datasets. *sa* - Saliency. *ig* - Integrated Gradient.

separate correct from wrong predictions. Figure 4 shows a general drop of performance of the explainers when the predictions do not match the true label. So, mixing wrong and correct nodes will necessarily reduce the scores. We also see that the gradient-based method Saliency is the only method able to explain the model logic when the predictions are wrong. This is not surprising as model-aware explainability methods focus on the model's internal working and will always explain the logic before the phenomenon. Therefore, all current papers that generate explanations when the model is not 100% accurate, are naturally biased towards gradient-based methods. This small study also encourages using Saliency to produce good explanations of a wrong GNN as it can also serve users to have an easier acceptance of bad models if they can actually explain them.

### 5.2.3 Select a pertinent explainability method

Based on the experiments, we outline how the design dimensions of GraphFramEx enables domain-specific users to quickly find best explanability models for their GNN prediction tasks. GraphFramEx finds the most appropriate method according to the 3 aspects described in section 4.1 and the accuracy level of the trained model, and can be shown as a decision tree. Figure 5 presents one decision tree when we set the mask transformation as the *topk* strategy with 10 edges ($k = 10$), for brevity purposes. It guides users to select the optimal method according to their multi-objectives and suggest explainers that are the best at returning necessary (red box), sufficient (green box) or both necessary and sufficient explanations (orange box). Other design considerations such as runtime can also be easily included based on the experiments. Note that additional explainability methods can be easily incorporated in our evaluation framework and be considered in the decision tree for general users.

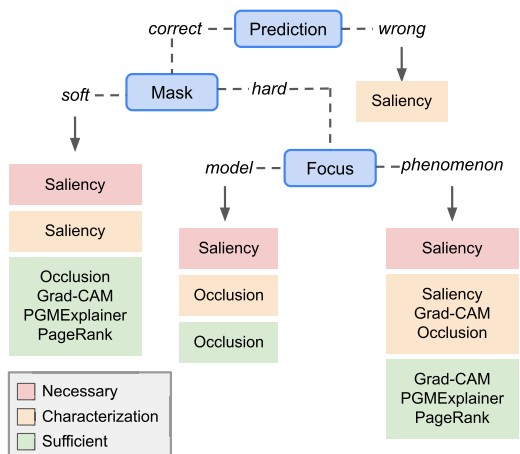

**Figure 5:** GraphFramEx decision tree for a mask transformation $topk = 10$.

### 5.2.4 Further Analysis

**Trade-off**. As observed in the two previous sections, Saliency seems to outperform the other methods except when we want sufficient explanations. In this case, Occlusion is the most appropriate one. We investigate if Saliency dominates the other methods. Figure 6 compares Saliency and Occlusion, respectively the first and second best methods on each dataset. Even though Saliency seems to dominate Occlusion to explain both model and phenomenon, we observe that it actually underperforms for Wisconsin, Actor and Facebook datasets when the focus is the model. We also observe that Occlusion is better at returning sufficient explanations, while Saliency is more appropriate for necessary explanations. This trade-off study shows that there is no existing explainability method that dominates others in all aspects.

**Limit of synthetic benchmarks**. We further reveal the limitations of evaluating explainability methods on type 1 synthetic datasets. We show inconsistency between the method rankings on real and those widely used synthetic datasets [10]. While PageRank returns the most accurate explanations (right table on Figure 8), and has the best time-performance trade-off and characterization score (see Figure 7) on synthetic data, this structure-based method is not able to highlight the important entities of real graphs (see Figure 3). In addition, Saliency has one of the lowest accuracies on every synthetic dataset, while it is the most optimal method to explain GCNs on real graphs (see Fig. 3). Method assessment on synthetic datasets eludes the power of gradient-based methods and their ability to extract decisive graph features when node dependency is not elementary and node features are meaningful. These examples demonstrate that evaluation on type 1 synthetic datasets gives only poor informative insight.

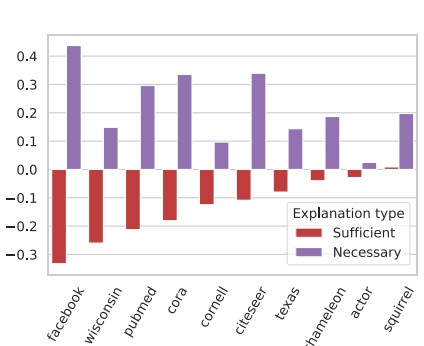
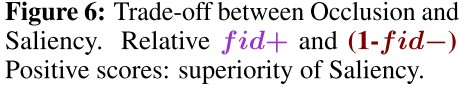

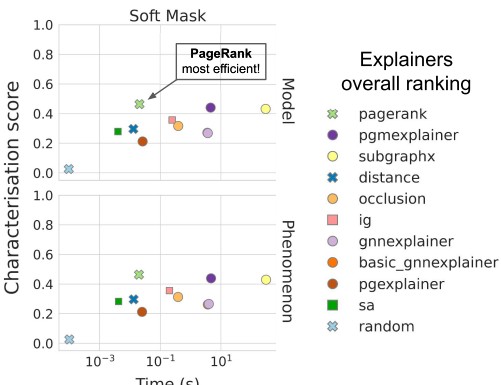

**Figure 6:** Trade-off between Occlusion and Saliency. Relative $fid+$ and $(1\text{-}fid-)$. Positive scores: superiority of Saliency.

**Figure 7:** Performance vs computation time for synthetic data. The explanation is a soft mask, *i.e.* edges are weighted by their importance.

| Explainer | BA-Shapes syn1 | BA-Grid syn2 | Tree-Cycle syn3 | Tree-Grid syn4 | BA-Bottles syn5 | CC-ratio |
|---|---|---|---|---|---|---|
| Random | 0.005 | 0.007 | 0.003 | 0 | 0 | 0.486 |
| Distance | 0.678 | 0.731 | 0.251 | 0.385 | **0.733** | **0.1** |
| PageRank | **0.862** | **0.938** | 0.323 | **0.485** | 0.658 | 0.128 |
| SA | 0.043 | 0 | 0.071 | 0.222 | 0.101 | 0.2 |
| IG | 0.447 | 0.186 | 0.244 | 0.435 | 0.400 | 0.18 |
| Occlusion | 0.172 | 0 | 0.220 | 0.268 | 0.074 | 0.206 |
| GNNExplainer | 0.310 | 0.664 | 0.233 | 0.328 | 0.127 | 0.176 |
| PGM-Explainer | 0.543 | 0.663 | **0.350** | 0.473 | 0.394 | 0.174 |
| PGExplainer | 0.652 | 0.801 | 0.204 | 0.236 | 0.455 | 0.262 |
| SubgraphX | 0.613 | 0.758 | 0.218 | 0.238 | 0.660 | 0.217 |

**Figure 8:** Accuracy on synthetic data. Explanations are generated to have the same number of edges than the expected groundtruth motif. (left) Explanatory subgraphs are drawn next to the expected ground truth. They contain the **target** node, **explanatory** nodes and **other** nodes. (right) F1-score indicates the similarity between the explanatory subgraph and the motif and CC-ratio the connectivity.

## 5.3 Case study: explaining frauds in the real-world e-commerce graph

We test our systematic evaluation framework on a production use case: explaining fraudulent transactions in the e-commerce scenario at eBay. In the scope of our research, we only explain correct predictions[1]. GNNexplainer is by far the most effective method (see Figure 9). It also returns not only sufficient explanations like most of the methods, but also necessary explanations. While the edge mask is directly deduced from the node feature mask in Saliency and Integrated Gradient, GNNExplainer has the particularity to compute edge and node features importance independently when solving the optimization problem. This explains the superiority of GNNExplainer in this production case where node features and edges bring both different insights to understand fraudulent nodes. Overall, we notice that perturbation-based methods are better than structure-based and gradient-based methods in this production use case.

## 6 Conclusion

In this paper, we propose GRAPHFRAMEX, a systematic evaluation framework for explainability methods for GNNs. By deliberately choosing methods from all categories, our comparison covers the full spectrum of input-level explainers for node classification tasks. Taking as model a GCN, we show the limits of a traditional evaluation on type 1 synthetic data. Our evaluation with the characterization score allows us to fairly evaluate all sorts of explainability methods in real-world scenarios. With our trade-off study, we however want to raise awareness that users should not rely on one single

---

[1]To circumvent the classification error of the trained GNN (Appendix B)

**Figure 9:** Results on eBay graph to explain correctly predicted fraudulent nodes. Results for the model focus are omitted as they correspond to the phenomenon. Explanation size is $topk = 10$. *sa* stands for Saliency; *ig* stands for Intergrated Gradient.

method to explain and trust their decision-making algorithm. Our case study on eBay graph shows the outstanding performance of GNNExplainer for explaining correctly predicted fraudulent nodes.

GRAPHFRAMEX is intended to help users navigate through the increasing number of explainability methods for GNNs. We encourage people to evaluate new explainability methods on real data and/or the 3 synthetic benchmarks [25] - *type 2* synthetic data - as they better reflect real-world complexity. While our work interprets explanations as positive weights masking the existing graph entities, we also aim at exploring new definitions that also involve non-adjacent pairs of nodes and assess the negative impact of edges and node features on the predicted outcomes.

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

# A    Background and foundational concepts

## A.1    Interpretability, explainability and transparency

There is a general misunderstanding of the terms explainability and interpretability. While interpretability is the common term in the philosophical literature, the scientific community prefers the term explainability. For this reason, we will only make use of terms that come from the same etymology as "explain". An explanation is the process (and its product) aiming at making something intelligible through the provision of structured information. Thus, the word explanation can be misleading as it refers to both the method and the result. Note that, for practical reasons, we explicitly use the term "method" to designate the method ("explainability method" or "explanation method") and the term "explanation" to describe the result of this method. As opposed to general explanations, scientific explanations answer only why-questions, where premises are always followed by a deduction. This does not mean that the explanation is unique: we often observe the existence of a large space of alternatives for the same question. Therefore, explanations need to take into consideration the social aspect of the process. Explainability of machine learning models has recently become a top-priority in AI, where it is often abbreviated as explainable Artificial Intelligence (xAI) or interpretable Machine Learning (iML). We adopt the first initialism here to stay as general as possible.

## A.2    GNN models and explanation quality

There are several variants of GNNs (graph convolutional networks (GCNs) [38], graph attention networks (GATs) [39], graph isomorphism networks (GINs) [40]), and they differ in their aggregation strategy. In this paper, we restrict our evaluation framework to methods that explain GCNs. We tested our framework on the simple GCN architecture proposed by [38]. Some papers [16, 21, 22, 24, 26, 45, 46] have tested their method for different GNN models and report their results for each one. To rigorously measure the robustness of explainers to the change of GNN model, the authors of [47] define the *consistency* metric. It measures how accuracy varies across different hyperparameters of a model or model architectures. When comparing explanations for different GNNs, those papers tackle the question: does the performance of an explainability method depend on our initial choice of the GNN architecture? In the scope of this paper, we only want to raise awareness on the potential importance of the GNN model on the generated explanations.

## A.3    Taxonomy of explainability methods for GNNs

Even if close in meaning, the definitions presented in this section are not to be confused with the ones introduced in [1] and [48].

**Input-level/Local vs Model-level/Global explanations**. An *input-level* or *example-level* or even *local* explanation identifies features in a given input that are important for its prediction. In contrast, *model-level* or *global* explanations are input-independent: they investigate what input graph patterns can lead to a certain GNN prediction without respect to any specific input example. They explain the general behavior of the model.

**Intrinsic explanations vs Post-hoc explanations**. *Intrinsic* explanations are produced for models that are self-understandable like linear regression and decision trees. No external method is required to explain their outcomes. *Post-hoc* explanations are brought up for models with higher complexity like neural networks, including GNNs, that do not presume any knowledge of the inner-workings or type of model at hand. In this case, an external method called explainability method is required to bring some clarity.

**Model-aware vs model-agnostic explanations**. Among post-hoc explanations, we have *model-aware* explanations and *model-agnostic* explanations. *Model-aware* methods look inside the model to extract information. They directly study the model parameters to reveal the relationships between the features in the input space and the output predictions. *Model-agnostic* explanations consider the model as a black-box. To infer what elements are important in the input, they perturb the input and study the changes in the output.

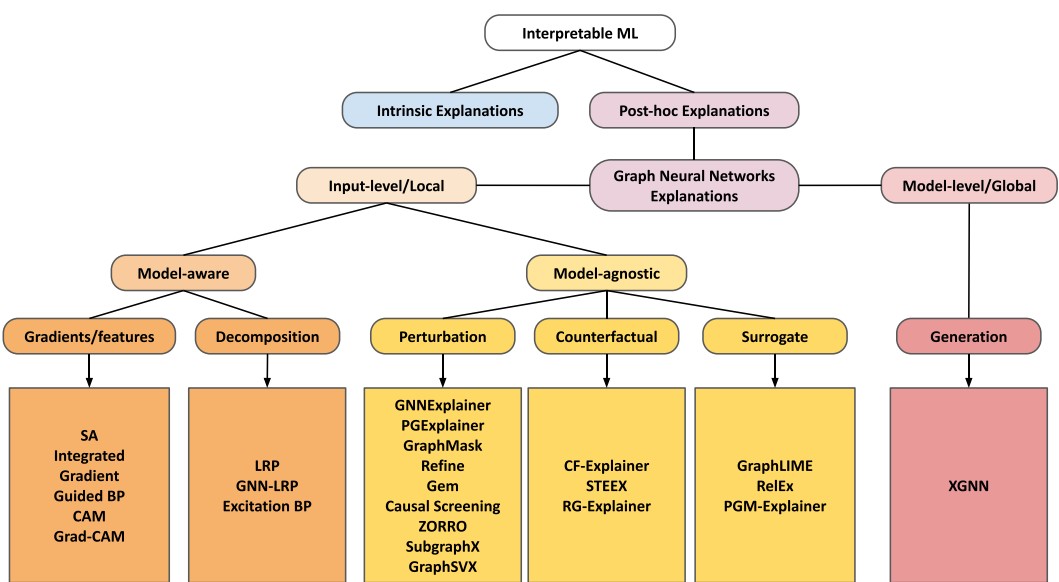

**Figure 10:** Explanation Taxonomy

## A.4 Fidelity measure

$$
\begin{array}{c|c}
\textbf{Phenomenon} & \textbf{Model} \\
\end{array}
$$

$$fid+^{acc} = \frac{1}{N}\sum_{i=1}^{N}\left|\mathbb{1}(\hat{y}_i = y_i) - \mathbb{1}(\hat{y}_i^{G_{C\setminus S}} = y_i)\right|$$

$$fid+^{acc} = \frac{1}{N}\sum_{i=1}^{N}\left|\mathbb{1}(\hat{y}_i = \hat{y}_i) - \mathbb{1}(\hat{y}_i^{G_{C\setminus S}} = \hat{y}_i)\right|$$

$$= 1 - \frac{1}{N}\sum_{i=1}^{N}\mathbb{1}(\hat{y}_i^{G_{C\setminus S}} = \hat{y}_i)$$

$$fid-^{acc} = \frac{1}{N}\sum_{i=1}^{N}\left|\mathbb{1}(\hat{y}_i = y_i) - \mathbb{1}(\hat{y}_i^{G_S} = y_i)\right|$$

$$fid-^{acc} = \frac{1}{N}\sum_{i=1}^{N}\left|\mathbb{1}(\hat{y}_i = \hat{y}_i) - \mathbb{1}(\hat{y}_i^{G_S} = \hat{y}_i)\right|$$

$$= 1 - \frac{1}{N}\sum_{i=1}^{N}\mathbb{1}(\hat{y}_i^{G_S} = \hat{y}_i)$$

$$fid+^{prob} = \frac{1}{N}\sum_{i=1}^{N}(f(G_C)_{y_i} - f(G_{C\setminus S})_{y_i})$$

$$fid+^{prob} = \frac{1}{N}\sum_{i=1}^{N}\left|f(G_C)_{\hat{y}_i} - f(G_{C\setminus S})_{\hat{y}_i}\right|$$

$$fid-^{prob} = \frac{1}{N}\sum_{i=1}^{N}(f(G_C)_{y_i} - f(G_S)_{y_i})$$

$$fid-^{prob} = \frac{1}{N}\sum_{i=1}^{N}\left|f(G_C)_{\hat{y}_i} - f(G_S)_{\hat{y}_i}\right|$$

We use the fidelity measures introduced in [26] to evaluate the contribution of the produced explanatory subgraph to the initial prediction, either by giving only the subgraph as input to the model (fidelity-) or by removing it from the entire graph and re-run the model (fidelity+). The fidelity scores capture how good an explainable model reproduces the natural phenomenon or the GNN model logic. The fidelity is measured with respect to the ground truth label or the predicted label according to

the focus choice. Equations A.4 detail the mathematical expressions of the different fidelity scores. The fidelity scores (+/-) can be expressed either with probabilities ($fid^{prob}_{+/-}$) or indicator functions ($fid^{acc}_{+/-}$). While $fid^{prob}_{+/-}$ metrics are more appropriate for evaluating explanations in the context of regression tasks because they are only based on the predicted probabilities, $fid^{acc}_{+/-}$ metrics use the indicator function and are more suitable for classification problems. In this paper, we convey our results with the fidelity metrics that use the indicator function and are more suitable for classification problems.

### A.5 Accuracy measure and the concept of groundtruth

The accuracy metric is based on the assumption that we actually know the groundtruth explanation. In current synthetic datasets, node labels are defined based on their position in the graph. Therefore, the groundtruth explanations are artificially built and interpreted as the motifs which the nodes belong to. We are critical towards this method of assigning explanations as it is an a posteriori assignment and is only based on the labeling procedure. How we, humans, synthetically build and explain the node labels is not necessary the right explanation of the GNN model logic. The GNN might put its attention on different graph entities than the ones of the human-intelligible substructures. For this reason, we claim here that accuracy is not the right evaluation metric as it is limited to datasets where we have ground-truth explanations and in these very rare cases, we strongly question their "ground-truth" quality.

### A.6 Classification of synthetic datasets

The term synthetic is widely used but its definition is not always clear. Synthetic refers here to data for which we have groudtruth explanations available. But, the procedure to generate the synthetic data and its groundtruth explanations differ. We have identified three origins of groudtruth:

- **Type 1 synthetic data** The true explanation is artificially defined by humans while they construct the graphs and can be identified as the nodes in the k-hop neighborhood of the target node. Such simple explanations can be easily discovered with nearest neighbor search or personalized PageRank. For instance, in the BA-house dataset, the motif house is the expected explanation. These synthetic datasets have been introduced in [10] and are now widely used as benchmarks to evaluate new explainability methods.
- **Type 2 synthetic data** The true explanation is also defined during the construction of the datasets. But, this time, it is more complex than the simple target node neighbourhood. Type 2 synthetic datasets correspond to the three benchmarks introduced in [25]. They have been created to overcome the 5 pitfalls encountered in type 1 synthetic datasets.
- **Type 3 synthetic data** The true explanation finds its origin in scientific experiments, human observations or human intuitions. Type 3 synthetic data often reflect biological and chemical problems, where particular substructures can predict properties for molecules, as in the MUTAG [29] or the MoleculeNet [30] datasets (HIV, BACE, BBBP, Tox21, QM7), or predict properties of proteins, as in the Enzymes dataset [29].

In this paper, we tested explainability methods on type 1 synthetic datasets to highlight their limitation in a rigorous evaluation of explainers. In addition, type 1 and type 3 are the most common families of synthetic data in recent papers [9–14, 16, 16–19, 26]. We have not tested the methods on type 3 synthetic datasets since they are made for graph classification and regression tasks.

### A.7 Mask transformation strategies

**Sparsity**. Sparsity is defined as the minimum percentage X of edges to remove from the initial graph. The sparsity strategy consists in keeping only edges which belong to the (100-X)% highest values in the mask. A sparsity of 70% or 0.7 means that we keep at least 30% of the edges in the mask. Some very sparse explainability methods might return sparser explanations with even less edges. But, we have the assurance that explanations cannot be bigger. Note that the size of the explanation is dependent on the size of the graph: if we change the dataset, the number of edges contained in the transformed masks will be different. Thus, for the sparsity strategy, the size of the explanation depends on the dataset.

| Datasets | | BA-House | BA-Grid | Tree-Cycle | Tree-Grid | BA-Bottle |
|---|---|---|---|---|---|---|
| Base | Type | BA graph | BA graph | Tree | Tree | BA graph |
| | Size | 300 nodes | 300 nodes | height 8 | height 8 | 300 nodes |
| Motif | Type | house | grid | cycle | grid | bottle |
| | Size | 5 nodes | 9 nodes | 6 nodes | 9 nodes | 5 nodes |
| | Number | 80 | 80 | 60 | 80 | 80 |
| # Features | | constant | constant | constant | constant | constant |
| # Classes | | 4 | 2 | 2 | 2 | 4 |

**Table 2:** Synthetic datasets statistics

| Datasets | Cora | CiteSeer | PubMed | Chameleon | Squirrel | Actor | Facebook | Cornell | Texas | Wisconsin |
|---|---|---|---|---|---|---|---|---|---|---|
| # Nodes | 2708 | 3327 | 19717 | 2277 | 5201 | 7600 | 22470 | 183 | 183 | 251 |
| # Edges | 5429 | 4732 | 44338 | 36101 | 217073 | 33544 | 171002 | 295 | 309 | 499 |
| # Features | 1433 | 3703 | 500 | 2325 | 2089 | 931 | 4714 | 1703 | 1703 | 1703 |
| # Classes | 7 | 6 | 3 | 5 | 5 | 5 | 4 | 5 | 5 | 5 |

**Table 3:** Real datasets statistics

**Threshold**. Threshold is a value between 0 and 1 that defines the lowest value for edge importance.The threshold strategy consists in keeping the edges whose value in the mask is greater than the threshold. For a threshold $\tau \in [0, 1]$, we keep only values in the mask greater than $\tau$. This leads to explanations of different sizes among the explainability methods, since some methods might value edges high while other methods give to their most important edges values below 0.5. Thus, for the threshold strategy, the size of the explanation depends on the method.

**Topk**. Topk is the number of edges in the explanatory subgraph. The topk strategy only keeps the top k highest values in the mask. This strategy always returns explanations with a similar absolute size whatever the dataset and the method. We also define the directed topk strategy and the undirected topk strategy. While the first one keeps the top k directed edges, the second one avoids double counting of node-to-node connections and returns explanations with k connections, i.e. the explanation is an undirected subgraph of k edges.

# B   Experimental details

## B.1   Datasets

Details on how the synthetic datasets were constructed can be found in Table 2. Table 3 presents the structural properties of the real datasets. eBay graph characteristics are detailed in Table 4.

**Synthetic datasets**. We use type 1 synthetic datasets introduced in [10] (see Appendix A.6), which are widely used in the xAI litterature [9–14, 16, 16–19, 26]. We follow the code[2] of Vu et al. [13] to create the synthetic datasets. In these datasets, each input graph is a combination of a base graph and a set of motifs. Diverse motifs (house, cycle, grid, bottle) are plugged in on a base graph (Barabasi graph or tree). Nodes are labeled based on their position in the graph: they receive a label 0 if they are in the base graph and a non-zero label if they belong to a motif. For house and bottle, the position in the motif is also important. For grid and cycle, we only look if the node belongs to the shape. The ground-truth label of each node on a motif is determined based on its role in the motif. As the labels are determined based on the motif's structure, the explanation for the role's prediction of a node are the nodes in the same motif. Thus, the ground-truth explanation in these datasets are the nodes in the same motif as the target.

**Citation datasets**. We consider three citation network datasets: Citeseer, Cora and Pubmed[49]. The datasets contain sparse bag-of-words feature vectors for each document and a list of citation links

---

[2]https://github.com/vunhatminh/PGMExplainer/tree/master/PGM_Node/Generate_XA_Data

| Dataset | # Nodes | # Txn Nodes | # Edges | # Features | # Classes | # Positive label | Train/Val/Test split |
|---------|---------|-------------|---------|------------|-----------|------------------|---------------------|
| eBay | 288853 | 207749 | 1225808 | 114 | 2 | 3081 (1.48% of *txns*) | 0.75/0.15/0.1 |

**Table 4:** eBay graph statistics

between documents. Citation links are treated as (undirected) edges. Each document has a class label. For training, we only use 20 labels per class, but all feature vectors.

**Facebook**. This dataset is a page-page graph of verified Facebook sites. Nodes correspond to official Facebook pages, links to mutual edges between sites. Node features are extracted from the site descriptions. The task is multi-class classification of the site category.

**Wikipedia network**. Chameleon and squirrel are two page-page networks on specific topics in Wikipedia. In those datasets, nodes represent web pages and edges are mutual links between pages. And node features correspond to several informative nouns in the Wikipedia pages. We classify the nodes into five categories in terms of the number of the average monthly traffic of the web page.

**Actor co-occurrence network**. This dataset is the actor-only induced subgraph of the film-director-actor-writer network. Each node corresponds to an actor, and the edge between two nodes denotes co-occurrence on the same Wikipedia page. Node features correspond to some keywords in the Wikipedia pages. Nodes are classified into five categories in terms of words on the actor's Wikipedia.

**WebKB**. WebKB1 is a web page dataset collected from computer science departments of various universities by Carnegie Mellon University. We use the three subdatasets of it, Cornell, Texas, and Wisconsin, where nodes represent web pages, and edges are hyperlinks between them. Node features are the bag-of-words representation of web pages. The web pages are manually classified into the five categories, student, project, course, staff, and faculty.

**eBay**. We conducted a case study on a real-world dataset with collaboration with the eBay Risk Team. We construct a bipartite graph with 2 different kinds of nodes: transaction nodes (*txn*), which are what we want to predict as targets, and entity nodes, which are unique assets including buyer account, payment tokens, email, IP address, and shipping address, acting like a linkage medium to connect *txns* together. If a *txn* has relation with an entity, we put an edge between these two nodes. Two different *txns* will be linked to the same entity node if they are sharing the same entity, *e.g.* the same shipping address is used in the two *txns*. Each *txn* is labeled as legit or fraudulent, and carries features provided by eBay risk system. These features include the information of transaction itself and expert-designed features extracted from its neighbors such as user and email information. For the entity nodes, the feature vectors are filled with zero value. Our source data is sampled from e-commerce history transaction logs. To ensure the connectivity of the graph, we first sample some seed *txns* within certain period of time, and then expand 3 hop neighbors from these seeds, and at each hop, no more than 32 neighbors are picked. Then we collect all involved nodes. The final graph has a size of 288,853 nodes (includes 207,749 *txn* nodes) and 1,225,808 edges. Among the *txn* nodes, 3,081 are labeled as fraudulent. Each *txn* node has 114 features. The graph we are using is the same with *eBay-small* graph in paper xFraud [50]. The desensitization version data is available for legitimate, non-commercial usage after submitting the application [3]. According to our experience, user based features usually contribute more, and payment tokens are usually a stronger evidence of fraud propagation among other entities. For example, a transaction with large user behavior change may be caused by account takeover attack, and a transaction using a payment token which has been used in other proved fraudulent purchases are more likely to be malicious.

## B.2 Explainability methods

**Model-aware**. Gradient-based methods compute the gradients of target prediction with respect to input features by back-propagation. Features-based methods map the hidden features to the input space via interpolation to measure important scores. Decomposition methods measure the importance of input features by distributing the prediction scores to the input space in a back-propagation manner.

---

[3] https://github.com/eBay/xFraud

**Model-agnostic**. Perturbation-based methods use masking strategy in the input space to perturb the input. Surrogate models use node/edge dropping, BFS sampling and node feature perturbation. Counterfactual methods generate counterfactual explanations by searching for a close possible world using adversarial perturbation techniques [51].

| Explainer | Model-aware/agnostic | Target | Type | Flow |
|---|---|---|---|---|
| SA | Model-aware | N/E | Gradient | Backward |
| IG | Model-aware | N/E | Gradient | Backward |
| Grad-CAM | Model-aware | N | Gradient | Backward |
| Occlusion | Model-agnostic | N/E | Perturbation | Forward |
| GNNExplainer | Model-agnostic | N/E/NF | Perturbation | Forward |
| PGExplainer | Model-agnostic | N/E | Perturbation | Forward |
| PGM-Explainer | Model-agnostic | N/E | Perturbation | Forward |
| SubgraphX | Model-agnostic | N/E | Perturbation | Forward |
| PageRank | Model-agnostic | N | Baseline | - |
| Distance | Model-agnostic | N | Baseline | - |

**Table 5:** Explainability methods tested in the context of our evaluation framework.

### B.3    GNN training

For all datasets, we use Adam optimizer [52]. The graph convolution network (GCN) has 2 or 3 layers with 16, 20 or 32 units. We eventually apply regularization on the weights with a weight decay factor of 0.05 or 0.005. We also apply dropout for some datasets. We indicate all parameters for each family of datasets. For synthetic datasets and for Facebook dataset, we use a 0.8/0.15/0.1 train/val/test split. For the Planetoid datasets, we use the default split: 140/500/1000 for Cora, 120/500/1000 for CiteSeer and 60/500/1000 for PubMed. We use the default train/val/test split for all other real datasets, namely 0.48/0.32/0.2. We further describe the model accuracy, F1-score, precision and recall for synthetic and real datasets.

### B.4    Protocol

For each dataset, we first train a graph convolution network (GCN) as introduced by Kipf and Welling [38]. For synthetic datasets, we use the version implemented by Rex Ying [4] [10]. For real datasets, we use the original GCN implementation from Kipf [5]. We use the trained model to do predictions of node targets of a testing set. We test twelve explainability methods on the synthetic and real datasets. We select 100 testing nodes which label we want to explain. We run each experiment on 5 different seeds and present the average results. All computations were run on ETH Zurich internal clusters:

---

[4] https://github.com/RexYing/gnn-model-explainer
[5] https://github.com/tkipf/gcn

| Datasets | Syn | WebKB | Citat., Wiki Faceb., Actor | eBay |
|---|---|---|---|---|
| layers | 3 | 2 | 2 | 2 |
| hidden dim | 20 | 32 | 16 | 32 |
| epochs | 1000 | 400 | 200 | 500 |
| learning rate | 0.001 | 0.001 | 0.01 | 0.001 |
| weight decay | $5 \cdot 10^{-3}$ | $5 \cdot 10^{-3}$ | $5 \cdot 10^{-4}$ | $5 \cdot 10^{-4}$ |
| dropout | 0 | 0.2 | 0.5 | 0.5 |

**Table 6:** GNN model and training parameters

| Datasets | BA House | BA Grid | Tree Cycle | Tree Grid | BA Bottle |
|---|---|---|---|---|---|
| accuracy | 0.986 | 1 | 1 | 0.895 | 1 |
| F1-score | 0.976 | 1 | 1 | 0.897 | 1 |
| recall | 0.979 | 1 | 1 | 0.87 | 1 |
| precision | 0.972 | 1 | 1 | 0.925 | 1 |

**Table 7:** GNN testing accuracy on synthetic datasets

| Datasets | Cora | CiteSeer | PubMed | Chameleon | Squirrel | Actor | Facebook | Cornell | Texas | Wisconsin | eBay |
|---|---|---|---|---|---|---|---|---|---|---|---|
| accuracy | 0.803 | 0.676 | 0.779 | 0.632 | 0.376 | 0.286 | 0.926 | 0.532 | 0.511 | 0.535 | 0.953 |
| F1-score | 0.799 | 0.652 | 0.777 | 0.64 | 0.35 | 0.254 | 0.92 | 0.344 | 0.285 | 0.397 | 0.594 |
| recall | 0.817 | 0.652 | 0.782 | 0.634 | 0.373 | 0.249 | 0.918 | 0.339 | 0.277 | 0.406 | 0.7 |
| precision | 0.781 | 0.651 | 0.772 | 0.647 | 0.333 | 0.261 | 0.923 | 0.355 | 0.297 | 0.398 | 0.566 |

**Table 8:** GNN model accuracy on real datasets and production data (eBay graph)

the HPC clusters of ID SIS HPC (Euler Clusters). We use 2 GPUs. For the synthetic datasets, we only explain the nodes that belong to a noteworthy motif (house, grid, cycle or bottle). For eBay, we only explain fraudulent nodes. The returned explanations are in the form of a mask over the nodes/edges and/or the node features. For the methods that return node masks, we convert those into edge masks: we assign to an edge the average importance of the nodes it connects. We apply the topk mask transformation strategy to reduce the size of the masks to k edges. This allows a fair comparison of the explainability methods by enforcing a similar size to all explanations. We compare our methods for different explanations with 1, 5, 10, 15, 20, 25, 50 and 100 edges. We also consider $k = 10$ edges as being a decent size for an explanation as it is still large to have interesting insight but sparse enough to stay human-intelligible. We compare methods in four different settings defined as the combinations of the two possible focuses, *phenomenon* and *model*, and mask natures, *hard* or *soft* masks.

## C   Additional results

### C.1   Variance analysis

We report here the average and standard deviation of our experiments run over 5 random seeds. Figure 11 shows the characterization score of the explainability methods on the real datasets. We observe that the variance stays small for every method, thus proving the robustness of our evaluation.

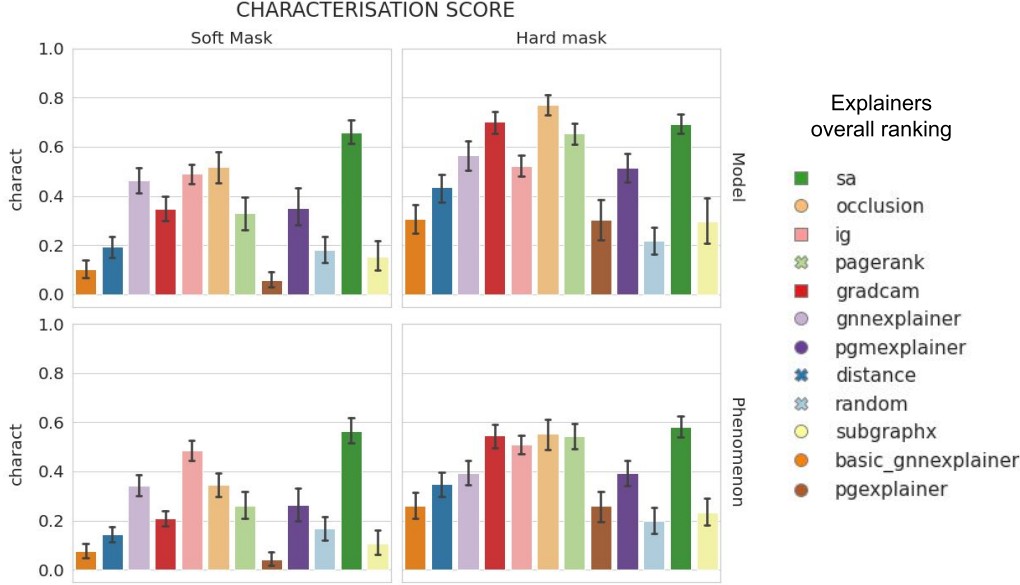

**Figure 11:** Characterization score of each explainability method in the four settings. Results are averaged over all real datasets and 5 different seeds and the standard deviation is indicated as the black segment.

### C.2   Mask properties and method performance

To understand the reasons behind the performance of an explainabiality method, we can further study the properties of its explanations. We look at four properties of the mask:

- **Mask size**: the number of edges selected by the method before any mask transformation. A large mask has non-zero values on most of the edges. A sparse mask has only a few non-zero edges.
- **Mask entropy**: the entropy of the value distribution in the mask. A high entropy means that the distribution of values in the mask is close to the uniform distribution: edges have different importance levels. A low entropy means that most of the values are concentrated around a few significant scalars: all edges have similar importance.
- **Mask maximum value**: the maximum importance level in the mask. A high maximum value means that the explainability method rates high important nodes. Note that if you have a large maximum value, i.e. close to 1, and the mask entropy is also large, the explanation covers the whole spectrum of importance levels.
- **Mask connectivity**: the fraction of connected components in the explanatory subgraph. The mask connectivity represents the ratio of connected components. Some explanations are composed of distinct edges without any connection, while others consist of one unique connected subgraph. The mask connectivity is computed as the fraction of number of connected components over the total number of edges in the explanatory subgraph. A ratio close to 0 indicates high connectivity in the explanation, while a ratio of 1 indicates that all edges are disconnected. We favor explanations with low ratios because highly connected explanations are more human-intelligible.

**Figure 12:** Explanation quality estimated with the characterization score vs. mask properties, *i.e.* entropy, maximum value and connected component ratio in a typical explanation of default size 10 edges. Results are averaged over all real datasets and 5 different seeds.

The mask analysis aims at finding correlations between methods' internal characteristics and their performance. We observe for instance that the best performing method Saliency generates masks with high entropy, i.e. the 10 edges have almost the same importance, high maximum value, i.e. all edges are considered as very important, and low connected component ratio, i.e. the explanation i almost one unique subgraph. When those three conditions are not met, it seems that the explanation cannot be a good characterization of the GNN predictions. This conclusion has been drawn from initial observations on some real datasets. Our study is still in its early stages. We have not yet found general rules that relate the mask structure to the method performance. In the future, we will further investigate if there exists a real correlation between the mask properties and the characterization score.

## C.3 Different GNN models

We have tested three GNN models: graph convolutional networks (GCN) [38], graph attention networks (GAT) [39] and graph isomorphism networks (GIN) [40]. GAT uses a neural network to learn the best weighting factors for aggregation and GIN's aggregator follows the Weisfeiler-Lehman test to better discriminate graphs. We report in Figure 13 the Fidelity+ and Fidelity- scores for the dataset Cora in the specific setting of explanations generated as hard masks for a phenomenon focus. We observe the strongest performance differences for the Fidelity- measure: Saliency's performance drops with GCN models, Integrated Gradients performs much better with GIN models, GNNexplainer seems to perfom well only with GAT models. The results are more consistent for Fidelity+. Based on these findings, we would like to further study the relationship between GNN models and epxlainability.

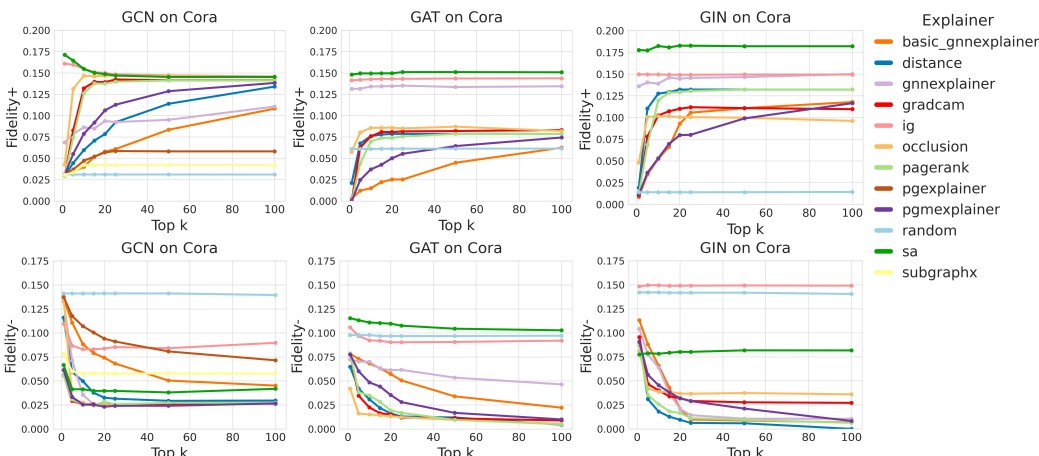

**Figure 13:** The Fidelity+ (top) and Fidelity- (bottom) comparisons between different GNN explanation techniques under different Topk levels for Cora dataset. Here, explanations are generated as hard masks (aspect 2) with a phenomenon focus (aspect 1).

## D Evaluation framework of explainability methods for graph classification

In this paper, we highlight the limitations of evaluation protocol of the explainability methods in the context of node classification. Here, we demonstrate to the readers that similar limitations exist for graph classification. Table 9 shows that existing works which seek to explain graph classification tasks only use small and different sets of metrics. In contrast with node classification, more papers test their methods on real graphs. But, they still do not consider the aspect of time and the influence of GNN accuracy on its explainability. Taking these observations into consideration, we wish to extend our evaluation framework to graph classification tasks in order to offer a universal evaluation protocol for any type of GNN tasks.

**Table 9:** XAI LITERATURE FOR GNN GRAPH CLASSIFICATION. Acc defines the accuracy (AUC, F1-score) measured with respect to the groundtruth, Fid+ and Fid- refer to the fidelity metrics as defined in [26]. "Time" indicates if the paper has run a comparative analysis of the computation time of the explainability methods. The final column "GNN accuracy" shows if the authors have reported the testing accuracy of their model.

| Paper Type | Year | Explainer | Target | Synthetic | | | Real | | | Time | GNN Accuracy |
|---|---|---|---|---|---|---|---|---|---|---|---|
| | | | | Acc | Fid- | Fid+ | Acc | Fid- | Fid+ | | |
| Method[9] | 2019 | LRP | E | | | | ✓ | | ✓ | | |
| Method[11] | 2020 | PGExplainer | E | | | | ✓ | | | ✓ | 0.92 - 1.00 |
| Method[12] | 2020 | RelEx | E | | | | ✓ | | | | |
| Method[13] | 2020 | PGM-Explainer | E | | | | ✓ | | | | 0.85-1.00 |
| Method[20] | 2020 | XGNN | E | | ✓* | | | ✓* | | | |
| Method[21] | 2021 | GNN-LRP | E | | ✓* | ✓* | | ✓* | ✓* | | 0.77-0.95 |
| Method[22] | 2021 | Causal Screening | E | | | | | ✓* | | ✓ | 0.64 - 0.98 |
| Method[16] | 2021 | SubgraphX | E | | | ✓ | | | ✓ | ✓ | 0.86-0.99 |
| Method[23] | 2021 | Refine | E | ✓ | ✓* | | ✓ | ✓* | | ✓ | 0.60-1.00 |
| Method[14] | 2021 | RG-Explainer | E | ✓ | | | ✓ | | | | |
| Method[19] | 2021 | Gem | E | | | | | ✓* | | ✓ | |
| Taxonomy[24] | 2019 | CG,EB,c-EB CAM,Grad-CAM | E | | | | | | ✓ | | 0.88-0.99 |
| Taxonomy[26] | 2020 | GNNExplainer,PGExplainer SubgraphX,DeepLift GNN-LRP,Grad-CAM,XGNN | E | ✓ | ✓ | ✓ | ✓ | ✓ | ✓ | | 0.44-0.91 |
| Taxonomy [28] | 2022 | VanillaGrad,IntergratedGrad GNNExplainer,PGMExplainer | E | | | | | ✓* | | | |

* Different denomination in the paper, but the same evaluation mechanism as ours.

