# OpenReview forum: "GraphFramEx: Towards Systematic Evaluation of Explainability Methods for Graph Neural Networks"
_logconference.io/LOG/2022/Conference — LoG 2022 Poster_

### Official Review · Reviewer_Kn2G · 2022-10-11

**Overall Score:** 8
**Confidence:** 4

**Review:**

This paper proposed a framework, namely GraphFramEx, to emulate the explainability of current explainable GNNs. The proposed GraphFrameEx splits the evaluation process into three steps, and a new emulation metric is constructed in the evaluation process. The motivation of the article is clear and the organization is reasonable, however, there are still some technique problems still exist. Therefore, it is suggested to consider a clear accept. My other questions are as follows:

[1]	The general protocol is shown in Fig. 1, however, it is hard to follow the contents shown in this figure. Does it need to preprocess the GNN model before estimation? In step 2, how the subgraph is obtained? Does it is obtained by the Mask generation process? Why the outputs of the model with subgraph input are the target? And how the authors link the results of steps 3 with step 2 to obtain the evaluation results?
[2]	Besides, in the step 3 “mask transformation”, how the transformation process will influence the evaluation results, the authors should give more explanations.
[3]	The results shown in Fig.3, Fig.4, Fig, 7 and Fig. 9 are hard to understand each subplot should have its own number and header.
[4]	I see the authors say they have compared the proposed methods for different explanations with different edges. So how will the number of edges influence the explanation results?
[5]	The authors mentioned that most current explainable GNNs are examined on the synthetic datasets and the authors also exam the GNNs on real datasets. So my question is, are there any limitations or requirements for the real data referred to here? For example, in industry I collect real-time data generated on a network of multiple sensors, how should the proposed method be assessed whether it is still applicable?
[6]	In the paper, the authors test different explainable GNN models and give the best performing model. But as an evaluation model, I think the authors should also give some instructive suggestions on how to design explainable GNN models.

---

### Official Review · Reviewer_HP5V · 2022-10-13

**Overall Score:** 8
**Confidence:** 4

**Review:**

##########################################################################

Summary:

This paper addresses the important problem of graph explainer evaluation. The authors propose a framework for the systematic evaluation of GNN explainers. They also introduce the notions of sufficiency and necessity for GNN explanations and propose a single metric to capture both concepts.

##########################################################################

Reasons for score:

Overall, I vote for accepting. I found the work comprehensive and of potentially great use for the community given the online portal, and leaderboard. My only concern of significance is related to the sufficiency concept introduced and the way the authors propose to measure (see concerns below). Hopefully the authors can address my concern in the rebuttal period.

##########################################################################

Pros:
1. The problem introduction is clear and gives a comprehensive overview of the state of the GraphXAI field.

2. This paper address key points often ignored by previous frameworks evaluating explanations: computing time, and influence of accuracy of the GNN model.

3. The evaluation method does not require ground truth and can be applied to any graph dataset in practice.

4. Comprehensive evaluation on synthetic and real datasets enabling insights into the new metric introduced (characterization score). Additionally, their observation “that there is currently no optimal method that generates the best explanations, as being both necessary and sufficient, in all experimental settings and for all datasets” will help researchers see that explanation methods should be chosen based on experimental settings and datasets.

5. Online portal for the evaluation of explainers with existing and their proposed metrics, including a leaderboard. This will propel the development of GNN explainers

##########################################################################

Comments I would like addressed during the rebuttal period:

##########################################################################

Cons:
1. The authors position well their contribution within existing works. Regarding work [28] I am not entirely sure what the authors are trying to convey in line 88: “they only take a model focus and do not account for necessary explanations.” Specifically: (1) what does it mean that authors in [28] take a model focus? And (2) to my understanding, “necessity” in explanations is a concept introduced by the authors in this work, it seems unfair to mention it as a drawback of work [28] (it would be in fact a drawback of all previous works if it were to be mentioned as such, right?).
2. The introduction of “sufficient” and “necessary” explanations is an interesting and novel idea. Regarding necessary explanations, the authors say that “an explanation is necessary if the model prediction changes when you remove it from the initial graph”. I don’t think this definition of a necessary explanation is entirely accurate in causal terms. A necessary explanation should be one that, if any element (or subset of elements) of it is removed, leads to a change in the prediction (or, in other words, one in which each element is necessary, given the remaining elements). The authors consider an explanation as a whole, but an explanation is composed of elements (nodes/edges in a predicted explanatory subgraph, for example) and any subset of those elements could potentially be the “necessary” set. Further, if we consider that a necessary explanation will always be a subset of a sufficient explanation, maybe the fid+ score is not accurate/ is insufficient to measure necessity and it is only another proxy for sufficiency - I would imagine that if a sufficient explanation is removed from the graph G, the prediction would change (unless there is another disjoint sufficient explanation).
It would be useful to investigate the relationship between the predictions with the masked graph and the complement – is it so that if the explanation with the masked graph G_S is correct (it is sufficient), the explanation without it (with G_{C\S}) is incorrect (it is necessary as per fid+)? If this happens in most cases, I think it empirically confirms that fid+ is not an accuracy proxy for causal necessity, and the concept should be revised. The right plot of figure 3 shows a somewhat linear trend between (1-)fid- and fid+, it would be useful to explore this further (ideally with each dataset individually).

##########################################################################

Some minor comments:
1. In line 108, what does \mathbb{G} represent? Should it just be G?
2. In line 111-113, I understand there is a mask M_E for each node v_t, maybe clarify this
3. Small detail, but maybe the authors would want to use subindex “t” instead of “i” in lines 117-118 to be consistent with the subindex used for the edge mask (lines 111-113).
4. In lines 144-146: “We can also want to explain how the model and open the black box works” – should this be: “We can also want to explain how the model works and open the black box”?
5. In line 158, am I understanding correctly that “every positive values to 1” means all values greater than zero are converted to 1? Could I ask if there is a reason for this specific design choice?

---

### Official Review · Reviewer_wr8N · 2022-10-21

**Overall Score:** 8
**Confidence:** 5

**Review:**

The Explainability of graph learning is a hot topic in recent years. And most existing methods adopt small and synthetic datasets for evaluation.  Definitely, we need a framework to comprehensively compare existing methods from different perspectives and this submission fill in this gap.  This paper is significant to the learning of the graph field.  Here is a small suggestion to improve the quality.

1. a  recent survey on the quantitative evaluation of XAI should be included.
[1] Nauta, Meike, et al. "From anecdotal evidence to quantitative evaluation methods: A systematic review on evaluating explainable ai." arXiv preprint arXiv:2201.08164 (2022).

---

### Official Review · Reviewer_wJdP · 2022-10-21

**Overall Score:** 5
**Confidence:** 4

**Review:**

Summary:
The paper proposes a framework for the systematic evaluation of post-hoc GNN explainability techniques on node classification tasks. The framework differentiates between three different user needs. Moreover, the paper presents a new metric combining fidelity measures and classifies explanations as sufficient and/or necesary.

Strengths:
1. The author's attempt to address an important challenge of further developing the benchmarking the performance of GNN explainers.
2. The authors publicise a leaderboard which researchers' can submit their explainers to.
2. Figure 5 nicely visualises the recommended decision making process for selecting an explainer.

Weaknesses:
1. Limited novelty. It is clear in the Related Work section of the paper that a number of other papers addressing the benchmarking problem exist. While there a slight differences from each paper on an individual comparison, overall the difference from the existing body of work is only moderate.
2. In the Method section, the section describing 'Aspect 2' could benefit from mentioning that the user is provided with a visualisation rather than a simple mask. Moreover, a strong theoretical rational for the framework is missing.
3. The difference between sufficient and necessary explanations is unclear.
4. The scores reported are an average over a set of datasets. This masks a lot of data as maybe the scores for explainers strongly vary with the dataset. Moreover, confidence intervals are missing.
5. The findings of the case study and the genreal findings are inconsistent without an explanation being offered.

Questions:
1. You define the explanation of model phenomenon as revealing findings in the data rather than the logic of the model. Does the inner working of the model not also reveal findings in the data? I might suggest rewording.
2. Does explaining wrong predictions impact the scores to the same extent across datasets for a given explainer?

General Feedback:
1. The abstract could benefit from being shorter and more clear in the description of the proposed metric.
2. You mention 'our explainer' in the Problem Setup section. This makes it unclear whether you are proposing an explainer.
3. There are a few typos. For example, the incorrect use of to vs on or an unmatched closing bracket in the Introduction section.
4. The register, tone and writing style change throughout the paper and should be standardised.
5. Check the consistency of using numeric citations and the author's name when referring to a paper directly. (e.g. [23] propose ... vs Bob et al. [23] propose ...)
6. Figure 1 is a little hard to follow.
7. The wording 'can want' should be revised.
8. Report confidence intervals and error margins, especially when averaging over a range of dataset.

Recommendation:
Weak Reject

Reasons for Recommendation:
I have given the paper a weak reject due to a 3 key reasons. Firstly, the writing style of the paper could be improved for clearer communication of the ideas. Secondly, the paper lacks a rigorous evaluation. While a number of metrics are investigated, important markers such as confidence intervals are missing. Findings could also be grounded by relating them to how the explainers work under the hood. Lastly, the contribution of the paper is somewhat limited as there is overlap with existing work, however, the flowchart on how to pick an explainer is done nicely!

---

### Meta-Review · Area_Chair_gcyE · 2022-11-17

**Confidence:** 4
**Recommendation:** Reject

**Meta Review:**

The paper aims to develop an evaluation method for GNN explanations. Extensive discussions are conducted among reviewers and authors. The remaining unresolved questions include: a more clear conclusion would better be provided in the paper as to which explainer is more desirable, the correlation between explainability and input data should be better clarified, better justify the exact definition of sufficiency and necessity as well as their relationship and difference with existing literature, better justify how and why Fid+ and Fid- are good metrics for sufficiency and necessity, and whether the differences among the methods to better understand the specificity of their explanations will be included in the manuscript.

---

### Decision · Program_Chairs · 2022-11-23

**Decision:**

Accept (Poster)

**Comment:**

PCs discussed this paper. After carefully looking over the reviews and the rebuttal, we find that this paper is fit for publication at the conference.